# Polymer-Dispersed Cholesteric Liquid Crystal under Homeotropic Anchoring: Electrically Induced Structures with *λ*^1/2^-Disclination

**DOI:** 10.3390/polym14071454

**Published:** 2022-04-02

**Authors:** Anna P. Gardymova, Mikhail N. Krakhalev, Vladimir Yu. Rudyak, Vadim A. Barbashov, Victor Ya. Zyryanov

**Affiliations:** 1Kirensky Institute of Physics, Federal Research Center KSC SB RAS, Krasnoyarsk 660036, Russia; kmn@iph.krasn.ru (M.N.K.); zyr@iph.krasn.ru (V.Y.Z.); 2Institute of Engineering Physics and Radio Electronics, Siberian Federal University, Krasnoyarsk 660041, Russia; 3Faculty of Physics, Lomonosov Moscow State University, Moscow 119991, Russia; vurdizm@gmail.com; 4Lebedev Physical Institute of the Russian Academy of Sciences, Moscow 119991, Russia; vadbar13@lebedev.ru

**Keywords:** polymer-dispersed cholesteric liquid crystal, orientational structure, electric field-induced transformation, topological defect, metastable state

## Abstract

Orientational structures of polymer-dispersed cholesteric liquid crystal under homeotropic anchoring and their transformations under the action of an electric field are studied. The switching of cholesteric droplets between different topological states are experimentally and theoretically demonstrated. Structures with $\lambda^{+1/2}$-disclination are found and considered. These structures are formed during the transformation of a twisted toroidal configuration induced by a decrease in the electric field when a relative chiral parameter $N_{0}>6.3$. The transformation of the initial structure with a bipolar distribution of the helix axis into a twisted toroidal configuration and then into a structure with $\lambda^{+1/2}$-disclination is investigated in detail. The behavior of these structures under the influence of an external electric field, as well as the appearance of structures with $\lambda^{-1/2}$-disclination, are studied. Obtained results are promising for the development of optical materials with programmable properties.

## 1. Introduction

Dispersions of cholesteric liquid crystal (CLC) droplets are used to develop various smart materials and find practical use as controllable reflective colour coatings [1,2,3,4,5], programmable lasers [6,7,8,9], bio- and chemosensors [10,11,12,13,14,15], etc. The application multiplicity of CLC dispersions is due to the vast diversity of the formed orientational structures, which are determined by the boundary conditions, the droplet’s shape, the CLC elasticity constants, the ratio of the droplet’s size *d* and the cholesteric intrinsic helix pitch $p_{0}$ characterized by the relative chiral parameter $N_{0}=2d/p_{0}$ [16,17,18,19,20,21]. For example, luminescence or Bragg reflection of light by cholesteric depends on both the CLC pitch [22,23,24,25] and the orientation of the cholesteric layers in the droplet [3,26,27].

Various stable and metastable states can be realized in CLC droplets under homeotropic boundary conditions (the liquid crystal director is oriented perpendicular to the interface) [28]. For instance, orientational structures with a different number and type of bulk point defects or structures with the linear surface defect are formed in CLC droplets at $N_{0}<6$ [20,29,30]. The cholesteric tends to form a pronounced layered configuration (layer-like structures) in droplets at great $N_{0}$ values [18,31]. In this case, droplets of a cholesteric polymer dispersed in an isotropic polymer can form up to six various layer-like structures [32]. For low molecular weight CLC, a smaller number of various layer-like configurations have been described: The structure with a bipolar distribution of the helix axis [33,34,35], the structure of nested cups [33,36], and structure with non-uniform cholesteric order in bulk [18].

It is known that a cholesteric can be switched from a stable state to a metastable state with a long lifetime by external influences. For example, switching between Grandjean planar, focal conic, and uniform lying helix structures is possible in a flat CLC cell [37]. Similar switchings between stable and metastable states are possible for CLC droplets, but such transitions and the resulting metastable configurations for the cholesteric droplets under homeotropic boundary conditions have not been studied thoroughly. For example, in Ref. [35] it was shown that the structure with a bipolar distribution of the helix axis unwinds into a nematic-like configuration under an electric field’s action, and a new metastable CLC structure formed at abruptly switched-off applied voltage. However, the resulting new configuration was not studied in detail.

In this paper, we present the results of studying the structures with $\lambda^{\pm 1/2}$-disclination formed in polymer-dispersed CLC droplets under homeotropic boundary conditions at a gradual decrease in the applied electric voltage; the paper also describes their formation dynamics and the response to the applied electric field.

## 2. Materials and Methods

### 2.1. Experimental Approaches

The cholesteric liquid crystal under study was the mixture of nematic E7 (Merck) and chiral dopant cholesteryl acetate (Sigma Aldrich, St. Louis, MO, USA). The chiral dopant concentration was 3 wt % corresponding to the intrinsic helix pitch $p_{0}=5.5\;\mu$m. CLC was dispersed in poly(isobutyl methacrylate) (PiBMA) (Sigma Aldrich) polymer specifying homeotropic anchoring for E7 nematic. The electro-optical sandwich-like cells were manufactured by combining the solvent-induced and thermal-induced phase separation techniques as described in Ref. [38] in detail. The weight ratio of CLC and PiBMA was CLC:PiBMA = 60:40, the composite film thickness *H* was specified by 20 μm diameter glass microspheres (Duke Scientific Corporation, Fremont, CA, USA).The experimental investigations were conducted using Axio Imager.A1m (Carl Zeiss, Oberkochen, Germany) polarization optical microscope (POM) at T=24°C. AC electric field (1 kHz) was applied perpendicular to the sample film. The voltage (the electric field) amplitude was increased with the 1 V RMS (0.05 V/μm) step. The time interval between successive voltage increases was equal to 1 min. The voltage was decreased step by step in a similar way.

### 2.2. Computer Simulations

#### 2.2.1. Calculations of the Droplet Structure

We considered oblate ellipsoidal droplets of dimensions d×d×h and the compression ratio δ=h/d=0.71. On the surface, strong homeotropic boundaries were set. We used the extended Frank elastic continuum approach with Monte-Carlo annealing optimization [39] on a 48×48×34 lattice. The energy of the system consisted of the director field distortion energy, the formation of defects and the interaction with an external electric field:(1)$$\begin{array}{c} F=\int_{V}\left(\frac{K_{11}}{2}{\left(\mathrm{div}n\right)}^{2}+\frac{K_{22}}{2}{\left(n\cdot \mathrm{rot}n+q_{0}\right)}^{2}+\frac{K_{33}}{2}{\left[n\times \mathrm{rot}n\right]}^{2}\right)dV\\ +\frac{W}{2}\int_{\Omega }\left(1-\cos^{2}\gamma \right)d\Omega +F_{def}+\varepsilon_{0}\Delta \varepsilon \int_{V}\left(E^{2}-{\left(E\cdot n\right)}^{2}\right)dV. \end{array}$$

Here $K_{11}=1$, $K_{22}=0.685$ and $K_{33}=1.54$ are the splay, twist and bend elasticity constants, respectively, $q_{0}=2\pi /p_{0}$, $W=600K_{11}/d$ is the surface anchoring energy density, γ is the angle between local director and normal to the droplet surface, $F_{def}$ is the energy of defects, Δε is dielectric anisotropy of LC, and E=(0,0,E) is the electric field. The types, positions and energies of defects were estimated automatically during the Monte-Carlo optimization procedure (see details in Ref. [39]).

#### 2.2.2. Calculation of the POM Textures

POM textures were calculated by the formulation of Jones matrices [40]. The resulting image was merged from calculations of the individual wavelengths of visible spectrum (from 400 nm to 700 nm) with their weights according to ≈3000 K black body luminescence spectra. Light diffraction, diffusion and scattering were not taken into account. The values of ordinary and extraordinary reflective indices for E7 cholesteric were set according to [41].

## 3. Results and Discussion

### 3.1. Transformation of the Initial Structure with a Bipolar Distribution of the Helix Axis

The response and the following relaxation of orientational structures to an electric field were studied concerning the CLC droplets with a relative chiral parameter of $5.5<N_{0}<11$. Initially, a structure with a bipolar distribution of the helix axis was formed in all the droplets under study (Figure 1a and Figure 2a) [35]. A low applied voltage leads to the orientation of the droplets’ bipolar axis perpendicular to the electric field direction. An increase in voltage leads to the gradual structure unwinding, accompanied by the cholesteric layers shifting to the droplet poles and their subsequent vanishing (Figure 1b). The physical origin of this process is similar to the unwinding of cholesteric layers in the bulk: With increasing voltage, the energy of interaction with the electric dominates on the twist energy. As a result, an equilibrium pitch increases, and the narrow π twist areas occur. Finally, at a critical voltage, the equilibrium pitch grows to infinity, and the structure comes to the twisted toroidal configuration with the symmetry axis oriented along the applied field being realized at high voltage (Figure 1c and Figure 2b) [38]. The relaxation process of CLC droplets and the type of structure formed with the decrease in voltage depends on the $N_{0}$ value.

The twisted toroidal configuration can remain in droplets at $N_{0}\leq 6.3$ when the voltage decreases to $U_{reduc.}=0$ V. The applied voltage value determines the central region size, where the director is oriented predominantly parallel to the electric field (Figure 2b,c) [38].

The observed experimental data conform to the numerical simulation results of the CLC structure. The stable state with the lowest energy corresponds to a structure with a bipolar distribution of the helix axis (the initial structure). The angle between the droplet bipolar axis and the short spheroid axis is approximately $50^{\circ }$ (Figure 1a). A gradual unwinding of the structure leading to a twisted toroidal configuration formation occurs under an electric field’s actions. The axisymmetric twisted toroidal structure is retained with a reverse decrease in the applied field (Figure 1d).

The twisted toroidal structure becomes unstable at $U_{reduc.}\geq 0$ V in CLC droplets at $N_{0}>6.3$. As a result, the twisted toroidal structure transforms to a layer-like configuration differing from the initial one when the field decreases to some critical value (Figure 2d). Lines corresponding to the orientation of the director perpendicular to the sample plane are observed in POM photos of layer-like structures, which makes it possible to determine the cholesteric layers’ orientation in the droplet’s central cross-section [35]. In particular, the cholesteric layers at the structure with a bipolar distribution of the helix axis are the circle arcs whose centers lie on the bipolar axis (Figure 2a). The new layer-like configuration is characterized by mirror symmetry of cholesteric layers arrangement and a $180^{\circ }$ bending of the one cholesteric layer (Figure 2d). This cholesteric layers arrangement is similar to a cholesteric structure with a $\lambda^{+1/2}$-disclination forming in flat CLC cells [42]. Additionally, two cholesteric layers identical to the layers in the original structure with a bipolar distribution of the helix axis are observed near the droplet border in the direction perpendicular to the symmetry plane. The configuration with $\lambda^{+1/2}$-disclination is stable and kept with voltage reduction to $U_{reduc.}=0$ V (Figure 2d).

Figure 3a,b shows in more detail a new layer-like structure calculated at $N_{0}=7.0$. The angle between the short droplet axis and the normal to the cholesteric layers equals approximately $50^{\circ }$ in the droplet’s central cross-section. The director distributions in the cross-section perpendicular to the cholesteric layers (by the plane (2)) (Figure 3a (top and middle rows)) and the cross-section by the plane (3) (Figure 3a (bottom row)) show that the arrangement of cholesteric layers forms a $\lambda^{+1/2}$-disclination. The linear surface defect has a characteristic pattern clearly revealed when the structure is observed in the direction perpendicular to the $\lambda^{+1/2}$-disclination (Figure 3 (bottom line)). In addition to the central layers forming the $\lambda^{+1/2}$-disclination, additional layers characteristic of a structure with a bipolar distribution of the helicoid axis are observed. The number of such cholesteric layers depends on the $N_{0}$ value (Figure 3b–e).

### 3.2. The Value of the Critical Electric Field

At a gradual decrease in the applied electrical field, the transformation of the twisted toroidal configuration into the structure with $\lambda^{+1/2}$-disclination occurs at the $N_{0}$ value-dependent critical electric field. Generally, the greater value of $N_{0}$ corresponds to the higher critical voltage (Figure 4). In the droplets at $N_{0}<11$, the twisted toroidal structure is kept when the applied electric field decreases to $E_{reduc.}=0.30$ V/μm. At the decreasing electric field to $E_{reduc.}=0.25$ V/μm, the structure with $\lambda^{+1/2}$-disclination is formed in some of the droplets at $8.1\leq N_{0}<11$, while the rest of the droplets keep the twisted toroidal configuration. A subsequent decrease in electric field to $E_{reduc.}=0.20$ V/μm leads to forming the layer-like structure in all droplets at $9.2\leq N_{0}<11$ and in some of the droplets at $7.1\leq N_{0}<9.2$. At $E_{reduc.}=0.15$ V/μm, the layer-like configuration is observed in droplets at $7.3\leq N_{0}<11.0$, and the twisted toroidal configuration is retained in droplets at $N_{0}<6.6$. At a subsequent decrease in electric field to $E_{reduc.}=0.10$ V/μm, both structures are observed in droplets at $6.3<N_{0}<6.4$. At $E_{reduc.}\leq 0.05$ V/μm, the structure with $\lambda^{+1/2}$-disclination is formed in droplets at $N_{0}>6.3$, and the twisted toroidal configuration is retained in all droplets at $N_{0}<5.9$.

### 3.3. Formation Dynamics of the Layer-like Structure with $\lambda^{+1/2}$-Disclination

A structure with $\lambda^{+1/2}$-disclination is formed during the twisted toroidal configuration transformation induced by a decrease in an electric field. To understand the formation mechanism of the layer-like structure, we studied the change dynamics of the configurations and demonstrated it using a CLC droplet at $N_{0}=7.7$ (Figure 5). Significant changes in the structure occur when the voltage decreases to $U_{reduc.}=4$ V. The maximum director rotation over the droplet diameter is limited to $180^{\circ }$ in a twisted toroidal structure [38]. As a result, at voltage $U_{reduc.}=4$ V, the region of additional director rotation arises and grows in droplet bulk, breaking the structure’s symmetry (Figure 5a, Appendix A). This structure distortion occupies a significant part of the droplet and takes the form of an elongated arc for about 1 s (Figure 5b). The polarizing optical microscope texture of the droplet becomes mirror symmetric. Later, the arc edges reach the droplet surface (the left border of the droplet in Figure 5c), after which a cholesteric layer is formed from the droplet border along the structure symmetry plane (Figure 5d–f). The formation of the first cholesteric layer leads additionally to the director twisting along the direction perpendicular to the structure symmetry plane and the formation of more cholesteric layers. This process occurs initially in the upper (Figure 5d,e) and after in the lower (Figure 5f,g) areas of the droplet shown in Figure 5. The orientational structure transformation process ends about 20 s after the voltage is decreased to $U_{reduc.}=4$ V.

The observed structure formation dynamics suggest the transformation process consists of two main stages. At the first stage, the additional director twist deformation occurs only in the droplet bulk without any violation of the boundary conditions or the formation of defects on the droplet’s surface (Figure 5a,b). At the second stage, the twist deformation of the director in bulk distorts the surface circular defect located at the droplet equator, which allows the system to form the cholesteric layer (Figure 5c–f). A director deformation that has appeared in the form of an elongated arc is retained and stabilized, resulting in the $\lambda^{+1/2}$-disclination formation. Other cholesteric layers appear later by the director twist from the first cholesteric layer, making it possible to involve a surface defect in this process. As a result, these cholesteric layers are circular arc shaped, and the linear surface defect acquires a characteristic pattern.

### 3.4. Electric Field-Induced Transformation of a Structure with $\lambda^{+1/2}$-Disclination

The applied electric field gradually decreases the number of cholesteric layers of the layer-like structures (see Figure 1a,b). This process begins with a length reduction of the 180${}^{\circ }$ folded central cholesteric layer in the structure with $\lambda^{+1/2}$-disclination. A noticeable change in the structure (the $\lambda^{+1/2}$-disclination position) is observed at a field strength of $E_{rise}>0.30$ V/μm for droplets at $N_{0}=7.7$ (Figure 6a,b). Most of the $\lambda^{+1/2}$-disclination displacement occurs at fields $0.30<E_{rise}<0.40$, while the position of other cholesteric layers changes insignificantly (Figure 6b–d). At $E_{rise}=0.40$ V/μm, the director is oriented predominantly along the field, and there are three sections with almost identical cholesteric layers near the droplet boundary (Figure 6d). Increasing the electric field to $E_{rise}=0.45$ V/μm results in the twisted toroidal configuration formation (Figure 6e), which returns to the structure with $\lambda^{+1/2}$-disclination at reverse with regards to the gradual decrease in voltage (Figure 6f).

The response of a layer-like structure with a bipolar distribution of the helix axis to an electric field and its relaxation at a subsequent decrease in voltage has a significant hysteresis. This makes it possible to change the droplet structure to another stable state with $\lambda^{+1/2}$-disclination (Figure 7). On the other hand, the structure with three cholesteric layers formed at increasing voltage (Figure 6d and Figure 7b) is not realised from an axisymmetric configuration with a decrease in voltage (see Figure 5). Accordingly, one can expect other stable/metastable structure formation in this case as well. In the case of the structure from Figure 6d, a gradual increase in the director twist will occur as the voltage decreases. Since there are already cholesteric layers, director twist increasing co-occurs in three directions perpendicular to the cholesteric layers without the formation of any additional distortions in the droplet bulk. As a result, a metastable layer-like structure with three groups of bent cholesteric layers forming a distortion in the droplet center is observed (Figure 7c). This cholesteric layer arrangement is similar to a cholesteric structure with a $\lambda^{-1/2}$-disclination forming in flat CLC cells [42].

## 4. Conclusions

The response and relaxation of polymer-dispersed cholesteric droplets with homeotropic boundary conditions to an applied electric field have been studied. CLC droplets at a relative chiral parameter of $5.5<N_{0}<11.0$ have been investigated; the electric field was changed step by step. The structure with a bipolar distribution of the helix axis is observed initially. The CLC structure unwinds under the action of the electric field, resulting in the axisymmetric twisted toroidal configuration formation. This axisymmetric structure is kept after the electric field is switched off for CLC droplets at $N_{0}\leq 5.9$. At $6.3<N_{0}<11.0$, the layer-like structure with $\lambda^{+1/2}$-disclination is formed when the voltage is turned off. A new layer-like structure is formed with a decrease in the electric field due to the axisymmetric structure transformation consisting of two main stages. At the first stage, the additional director field distortion occurs in the droplet bulk without violating the boundary conditions. At the second stage, the surface circular defect at the droplet equator is curved, forming the cholesteric layer. The structure with $\lambda^{+1/2}$-disclination is observed in most droplets at $6.3<N_{0}<11.0$ when voltage is decreased step by step (see Appendix A). The $N_{0}$ value determines the number of cholesteric layers in the form of the circle arcs.

Hysteresis is observed in the response of CLC droplets to an electric field, which made it possible to realize another scenario of structure transformation leading to the formation of a layer-like configuration with $\lambda^{-1/2}$-disclination. This configuration is formed when three cholesteric layers are already in the relaxing structure. In this case, the director twist increasing co-occurs in three directions perpendicular to the cholesteric layers, leading to the layer-like structure with three groups of the bent cholesteric layers. Since the relaxation process depends on the number of the cholesteric layers, the possibility of forming droplets with the structure with $\lambda^{-1/2}$-disclination is defined by the $N_{0}$ value in contrast to the structure with $\lambda^{+1/2}$-disclination (see Appendix A).

Thus, we implemented electrical switching CLC droplets from a structure with a bipolar distribution of the helix axis to the twisted toroidal configuration and the layer-like structure with $\lambda^{+1/2}$-disclination or $\lambda^{-1/2}$-disclination. The results demonstrate that the control of the orientational structure formation process can be used to obtain various stable CLC droplet states. Further development of this approach may make it possible to obtain optical materials with programmable properties for electro-optics devises, photonics, optomechanics, etc. Moreover, the proposed approach to the CLC structure formation is suitable for the two-step preparation of PDLC materials [43,44] with the possibility of polymer stabilization of various orientational structures.

## Figures and Tables

**Figure 1 polymers-14-01454-f001:**
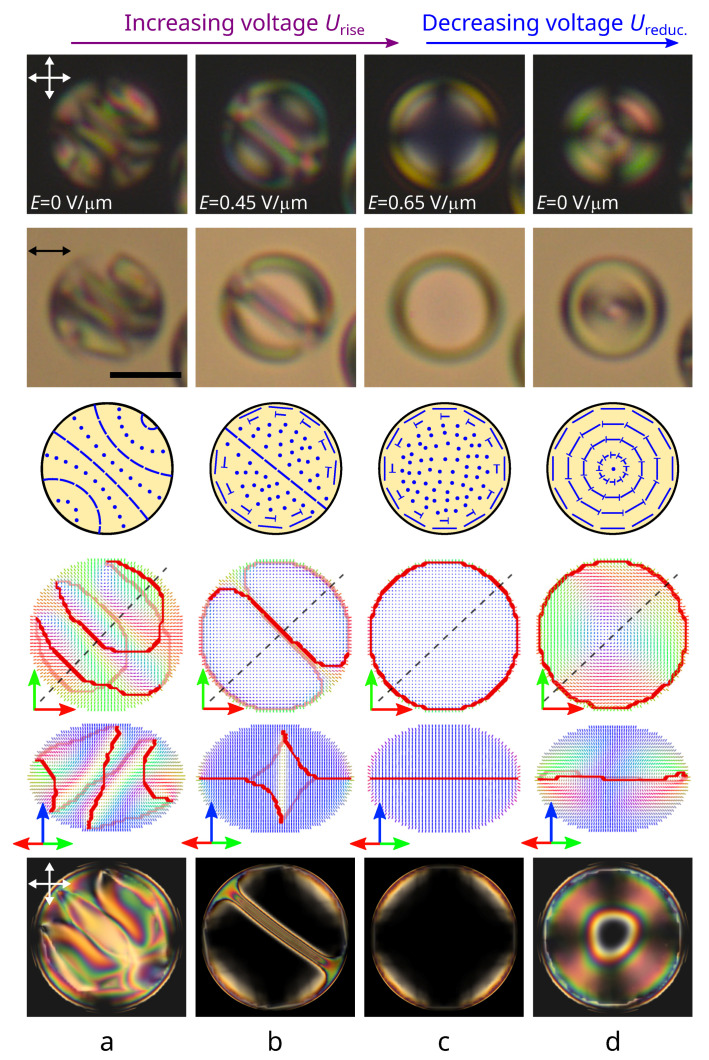
The initial $N_{0}=6.3$ structure with a bipolar distribution of the cholesteric axis (**a**), droplet under an electric field $E_{rise}$ equal to 0.45 V/μm (**b**) and 0.65 V/μm (**c**) when voltage is increased, and the droplet after an electric field is switched off (**d**). POM photos taken with the crossed polarizers (**top row**) and with one polarizer (**second row**); schemes of the director in the droplet cross-section (**third row**); calculated director distributions at the top view (**fourth row**) and side view (**fifth row**), and the corresponding POM pictures (**bottom row**). Hereinafter, double arrows indicate the directions of the polarizers, and the director **n** is colored in correspondence with the direction (red along the *x*-axis, green along the *y*-axis, blue along the *z*-axis). The scale bar is 10 μm.

**Figure 2 polymers-14-01454-f002:**
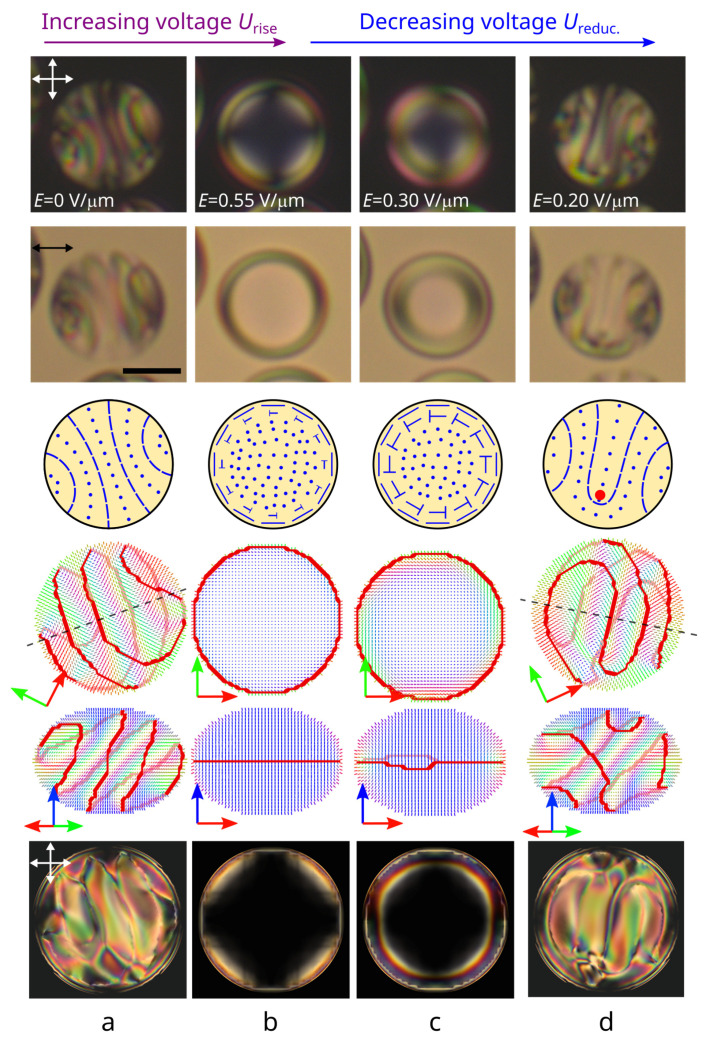
The initial $N_{0}=7.7$ structure with a bipolar distribution of the cholesteric axis (**a**), droplet under an electric field $E_{rise}$ equal to 0.55 V/μm (**b**) when voltage increases. The droplet under an electric field $E_{reduc.}$ equal to 0.30 V/μm (**c**) and 0.20 V/μm (**d**) when voltage decreases. POM photos taken with the crossed polarizers (**top row**) and with one polarizer (**second row**); schemes of the director in the droplet cross-section (**third row**); calculated director distributions at the top view (**fourth row**) and side view (**fifth row**), and the corresponding POM pictures (**bottom row**). The red circle indicates the cross-section of the $\lambda^{+1/2}$-disclination.

**Figure 3 polymers-14-01454-f003:**
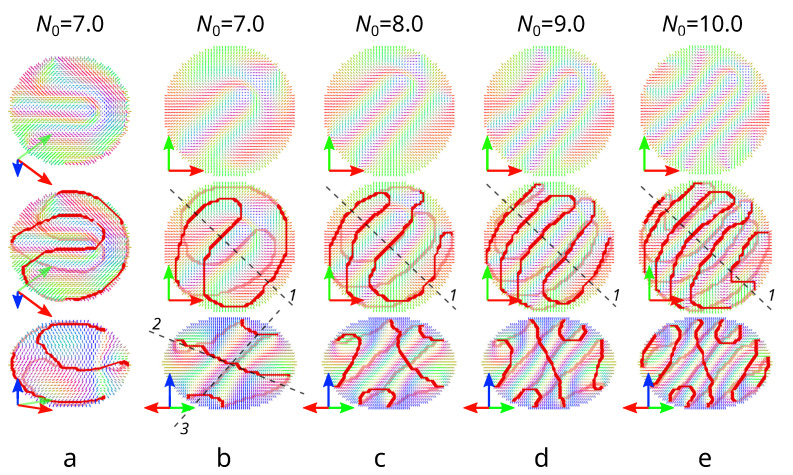
Director distributions in CLC droplets with $\lambda^{+1/2}$-disclination at $N_{0}=7.0$ (**a**,**b**), 8.0 (**c**), 9.0 (**d**) and 10.0 (**e**). (**a**) Director distributions in the plane (2) perpendicular to the $\lambda^{+1/2}$-disclination (**top and middle lines**) and in the plane (3) through the $\lambda^{+1/2}$-disclination (**bottom line**). (**b**–**e**) Director distributions in the equatorial plane—top view (**top and middle lines**); and in the plane (1) perpendicular to cholesteric layers—side view (**bottom line**). Red lines depict defect lines.

**Figure 4 polymers-14-01454-f004:**
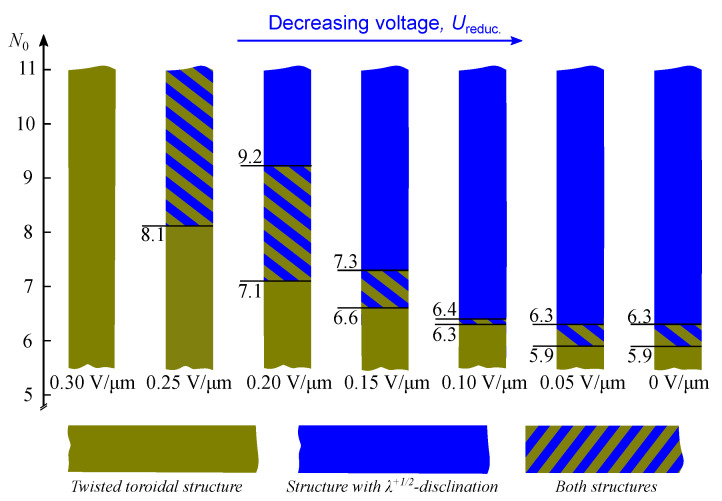
$N_{0}$ values at which a layer-like structure (**blue boxes**) and a twisted toroidal configuration (**olive boxes**) are observed, depending on an electric field $E_{reduc.}$. The olive-blue boxes indicate the $N_{0}$ values at which both structures are observed in CLC droplets.

**Figure 5 polymers-14-01454-f005:**
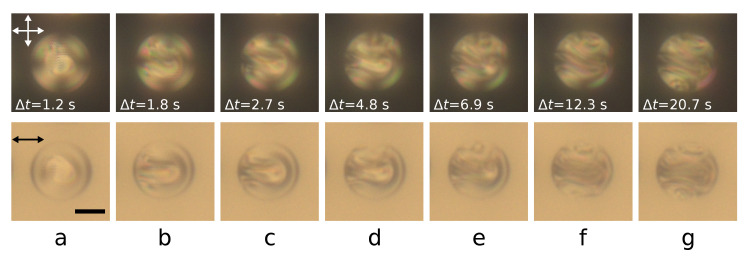
POM images of CLC droplet at $N_{0}=7.7$ taken with the crossed polarizers (**top row**) and with one polarizer (**bottom row**) Δt=1.2 s (**a**), 1.8 s (**b**), 2.7 (**c**), 4.8 s (**d**), 6.9 s (**e**), 12.3 (**f**) and 20.7 s (**g**) after the voltage was reduced from $U_{reduc.}=5$ V ($E_{reduc.}=0.25$ V/μm) to $U_{reduc.}=4$ V ($E_{reduc.}=0.20$ V/μm).

**Figure 6 polymers-14-01454-f006:**
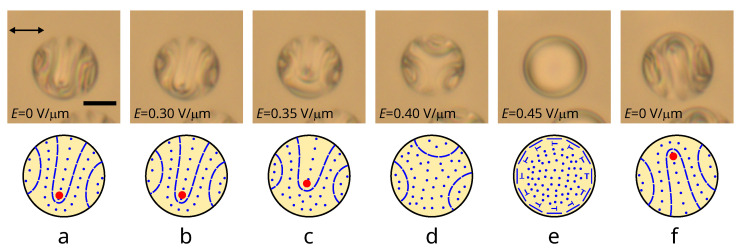
POM images of CLC droplet at $N_{0}=7.7$ taken with one polarizer (**top row**) and corresponding schemes of cholesteric layers (**bottom row**). CLC droplet before (**a**), under an electric field $E_{rise}$ equal to 0.30 V/μm (**b**), 0.35 V/μm (**c**), 0.40 V/μm (**d**), 0.45 V/μm (**e**) and after decreasing the electric field to $E_{reduc.}=0$ V/μm (**f**). The red circles indicate the cross-section of the $\lambda^{+1/2}$-disclination.

**Figure 7 polymers-14-01454-f007:**
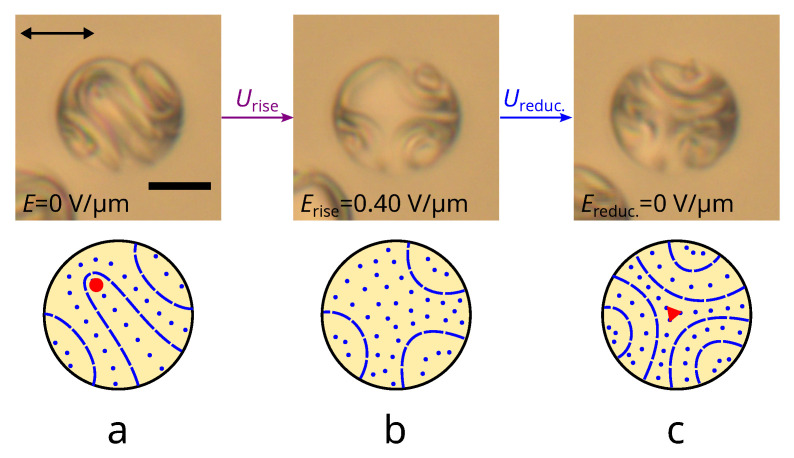
POM images of CLC droplet at $N_{0}=7.7$ taken with one polarizer (**top row**) and corresponding schemes of cholesteric layers (**bottom row**). The red circle and triangle indicate the cross-section of the $\lambda^{+1/2}$-disclination and $\lambda^{-1/2}$-disclination, respectively. CLC droplet before (**a**), under an electric field Erise equal to 0.40 V/m (**b**) and after decreasing the electric field to Ereduc. = 0 V/m (**c**).

## Data Availability

The data generated during the current study are available from the corresponding author on reasonable request.

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
