# Peer review of "Polymer-Dispersed Cholesteric Liquid Crystal under Homeotropic Anchoring: Electrically Induced Structures with λ1/2-Disclination"

_polymers, 2022, doi:10.3390/polym14071454_

Round 1

Reviewer 1 Report

The article by Gardymova et al. deals with polymer-dispersed cholesteric liquid crystal  under homeotropic boundary conditions. The authors have already published a lot of articles in the same topic. See for instance the following references of the article: 20, 21, 35, 38, and more that are not cited in the current article like “Untwisting of the helical structure of cholesteric droplets with homeotropic surface anchoring”. Therefore, I cannot find the novelty of the current work, and I cannot recommend it for publication

Reviewer 2 Report

Dear authors,

Thank you for submitting this manuscript. In this research, you systematically studied how PDCLC with homeotropic anchoring reacts with external electric field. I read your previous research (Polymers, 10.3390/polym13050732) and this manuscript, and find this research original and novel. I couldn't find obvious flaws in paper logic, experiment design and data analysis, so I'm happy to recommend a publication. I have some comments:

  1. In this research, the characterization approach is POM, and you compared the POM results of each experiment condition with computer modeling. Although this is enough to validate your experiment results, I recommend adding more discussion of physics behind the scene. For example, from Fig. 1b to 1c, is it possible to explain the change of director orientation by using free energy equation (1)?
  2. In the conclusion part, you propose this research may bring hope to programmable optical materials. If you could list a few potential applications of this research, the manuscript could be more attractive to readers.

Reviewer 3 Report

This paper describes the orientation structure of polymer-dispersed cholesteric liquid crystals (PDCLC) under homeotropic anchor conditions and its change with the application of an electric field, both experimentally and by simulation. PDCLC is a promising material for optoelectronic devices, and molecular orientation in the device is very important for the development of its properties. In this system, both theoretical simulations and experiments are consistent. This will be of interest to readers involved in device development. The results are worthy of publication in Polymers. Here are some points to consider.

1. In figures 1 and 2, it is worth to add the simulation results at maximum applied voltage (0.55 and 0.65 V/[micro]m)to compare between experimental and simulation.

Round 2

Reviewer 1 Report

As indicated in my former report, the authors have already publish many articles on PDLC, and although there are small variations between the papers, the work is not novel for publication